# Inclusion Formation and Toxicity of the ALS Protein RGNEF and Its Association with the Microtubule Network

**DOI:** 10.3390/ijms21165597

**Published:** 2020-08-05

**Authors:** Sonja E. Di Gregorio, Kathryn Volkening, Michael J. Strong, Martin L. Duennwald

**Affiliations:** 1Department of Pathology and Laboratory Medicine, University of Western Ontario, London, ON N6A 5C1, Canada; sdigreg@uwo.ca; 2Clinical Neurological Sciences, University of Western Ontario, London, ON N6A 5C1, Canada; kvolkening@robarts.ca (K.V.); michael.strong@schulich.uwo.ca (M.J.S.)

**Keywords:** RGNEF, ALS, yeast model, protein misfolding, microtubules, neurodegeneration

## Abstract

The Rho guanine nucleotide exchange factor (RGNEF) protein encoded by the *ARHGEF28* gene has been implicated in the neurodegenerative disease amyotrophic lateral sclerosis (ALS). Biochemical and pathological studies have shown that RGNEF is a component of the hallmark neuronal cytoplasmic inclusions in ALS-affected neurons. Additionally, a heterozygous mutation in *ARHGEF28* has been identified in a number of familial ALS (fALS) cases that may give rise to one of two truncated variants of the protein. Little is known about the normal biological function of RGNEF or how it contributes to ALS pathogenesis. To further explore RGNEF biology we have established and characterized a yeast model and characterized RGNEF expression in several mammalian cell lines. We demonstrate that RGNEF is toxic when overexpressed and forms inclusions. We also found that the fALS-associated mutation in *ARGHEF28* gives rise to an inclusion-forming and toxic protein. Additionally, through unbiased screening using the split-ubiquitin system, we have identified RGNEF-interacting proteins, including two ALS-associated proteins. Functional characterization of other RGNEF interactors identified in our screen suggest that RGNEF functions as a microtubule regulator. Our findings indicate that RGNEF misfolding and toxicity may cause impairment of the microtubule network and contribute to ALS pathogenesis.

## 1. Introduction

Amyotrophic lateral sclerosis (ALS) is a heterogeneous neurodegenerative disease caused by loss of upper and lower motor neurons in the brain and spinal cord [1]. ALS can be grouped into sporadic ALS (sALS), i.e., without any family history, which accounts for ~90% of all ALS cases, and familial ALS (fALS) with a family history of ALS, which accounts for ~10% of all ALS cases [2]. Despite considerable efforts, the molecular mechanisms underpinning the disease remain and there are no satisfactory treatments for ALS. Since the discovery of ALS mutations in *SOD1*, more than twenty-six genes have been identified as causative for fALS [3,4]. Many of these proteins are involved in RNA metabolism, including C9orf72, Tar DNA binding protein (TDP-43), and fused in sarcoma (FUS) [3,4].

Protein misfolding is a global hallmark of neurodegenerative disorders, including ALS [5,6,7]. A protein is considered to be misfolded when its soluble, native three-dimensional conformation is compromised leading to aberrant changes in structure and function. Misfolded ALS proteins, such as TDP-43, FUS, and SOD1, often mislocalize and form pathological aggregates and inclusions in the cytoplasm of motor neurons termed neuronal cytoplasmic inclusions (NCIs) [8]. Genetic mutations can severely increase the propensity of ALS proteins to misfold as demonstrated for FUS and TDP-43 and many other proteins [9,10,11]. However, most sALS cases are not linked to any known mutations. Environmental insults, such as changes in pH and exposure to toxic chemicals or excessive oxidative stress, can lead to protein misfolding that may contribute to neurodegeneration in ALS [12,13]. Finally, the highest risk factor for most neurodegenerative diseases is advanced age, indicating that the physiological changes associated with aging contribute to disease-related protein misfolding [14].

The Rho guanine nucleotide exchange factor (RGNEF) protein has recently been implicated in ALS [15,16,17]. RGNEF is a 191 kDa, RNA binding protein encoded by the *ARHGEF28* gene in humans. RGNEF belongs to the family of diffuse B-cell lymphoma (Dbl) GEFs and contains a leucine-rich region at its amino-terminus, a Plekstrin homology domain (PH), a cysteine-rich Zink-finger domain, FAK (focal adhesion kinase) binding domain, microtubule binding domain, and an nuclear localization signal (NLS, [18]. The normal cellular function of RGNEF is not well understood; however, it has been linked to multiple disorders in addition to ALS, including cancer, e.g., colorectal and ovarian cancer [19,20,21,22,23].

Biochemical and pathological studies implicate RGNEF as an ALS protein, yet how the mechanisms by which it contributes to ALS pathogenesis remain unknown. A heterozygous mutation in exon 6 of *ARHGEF28* has been identified in a small number of fALS cases [17]. These finding were supported by Ma et al., 2014, who detected the same RGNEF mutation in two additional ALS cases. Both these cases were heterozygous for the *ARHGEF28* mutations, and both identified patients showed similar ALS symptoms, including bulbar onset. These patients did not have a family history of ALS and did not possess overlapping mutations in either *SOD1* or *C9orf72* as the previously identified ALS patients [24]. The deletion of a single nucleotide is predicted to lead to either a frameshift mutation that gives rise to a premature stop codon resulting in a severely truncated RGNEF protein or may also cause splicing errors in exon 6 of *ARHGEF28* causing exon skipping that would generate a severely truncated RGNEF protein with a slightly different sequence from that of the first proposed mutation [17,25]. 

RGNEF has been found to localize to hallmark NCIs in ALS patient spinal cord motor neurons, where RGNEF co-localizes with other RNA binding ALS proteins such as TDP-43, FUS, and C9orf72 [16,26]. Additionally, RGNEF co-localizes with markers of protein degradation, such as p62 and ubiquitin [16,26]. The only variant of ALS that did not show RGNEF in NCIs tested so far were ALS cases bearing *SOD1* mutations.

Additionally, metabolic stress induces formation of micronuclei, small nuclear fragments in cultured cell models [27,28]. These structures are also found in the brains and spinal cords of ALS patients. TDP-43 inclusions in micronuclei co-localize with RGNEF and may be released into the cytoplasm [29]. The leucine-rich region of RGNEF is critical for interaction with TDP-43 and localization to micronuclei. In addition, RGNEF regulates neurofilaments via binding to the mRNA encoding low molecular weight neurofilaments. Human RGNEF destabilizes neurofilament mRNA and over expression of RGNEF reduced protein levels of neurofilament in stable cell lines [26].

In sum, all these findings document that RGNEF contributes to ALS pathogenesis, yet the precise underlying cellular mechanisms remain mostly unclear. Here, we established and characterized a yeast model and employed cultured neuronal cells to explore RGNEF misfolding, toxicity, and its interactions with major cellular pathways. Our results indicate that RGNEF can undergo toxic misfolding and that it interacts with the microtubule network, which may contribute to neurotoxicity and to neurodegeneration in ALS.

## 2. Results 

We first established and characterized an RGNEF yeast model. Figure 1A shows schematic representations of the domain structure of wild type and the truncated RGNEF variants used in our study. Upon expression in yeast, we found that the expression of full length wild type RGNEF reduced the growth of yeast cells by ~50% of control cells (*p* < 0.05, Figure 1B,C). Interestingly, the construct lacking the carboxyl-terminus, but containing the GEF domain of RGNEF (∆Cterm) was more toxic than the wild type RGNEF, reducing growth by over 50% (*p* < 0.05). Additionally, expression of construct lacking the amino-terminal leucine-rich region (∆Leu) is highly toxic in yeast, showing over 90% reduction of growth (*p* < 0.01) (Figure 1B,C). All other truncations tested here did not exhibit a growth defect in yeast. 

We monitored subcellular localization of RGNEF in yeast by expressing yellow fluorescent protein (YFP) carboxy-terminal fusion to full length RGNEF (RGNEF-YFP) and its truncated variants lacking specific functional domains (Figure 2A). All truncations and wild type RGNEF-YFP fusions form fluorescent foci in the cytoplasm of yeast cells, albeit to different proportions of cells (Figure 2B). Full length wild type RGNEF-YFP is found mostly localized in foci, however, not exclusively, with some cells display a fully diffuse signal, whereas other cells contain a single large or several small fluorescent foci. The bar graph in Figure 2B represents the proportion of cells containing large foci (black bar) as a percentage over the total number of cells. The construct lacking the carboxy-terminus (∆Cterm) showed less foci than wild type and the construct lacking the amino-terminal leucine-rich domain (∆Leu) showed a fraction of cells with foci similar to WT. Figure 2C documents full length wild type RGNEF expression by Western blot analysis.

Figure 3A shows schematic representations of the two ALS variants that are predicted to arise from mutations in *ARHGEF28* in fALS compared to the full length wild type protein and the leucine-rich region of RGNEF alone. We expressed in yeast these RGNEF variants, 259 and 319 named here after their respective amino acid lengths. Toxicity was observed when the larger truncation, 319, was expressed in yeast and to a lesser extent for the short variant, 259 (Figure 3B,C). Carboxy-terminally YFP-tagged constructs of 259 and 319 were utilized to examine the subcellular localization. When expressed in yeast we found that both mutants showed mostly diffuse signals with a small subset of cells showing a single large inclusion (Figure 3D,E).

Overexpression of RGNEF and the truncations lacking the carboxy terminus (∆Cterm) or the leucine-rich region (∆Leu) upon transient transfection of expression plasmids in HEK 293 cells corroborated the growth defects observed in yeast. Luciferase-based viability assays revealed toxicity in cells expressing the full length wild type RGNEF and ∆Cterm and ∆Leu as compared to vector only control (*p* < 0.05) (Figure 4A). Additionally, localization of full length RGNEF and ∆Cterm and ∆Leu in HEK 293 cells was evaluated using carboxy-terminally GFP-tagged fusions. Fluorescence microscopy revealed a mostly diffuse localization of wild type RGNEF, ∆Cterm, and ∆Leu expressing cells with a small proportion of cells containing fluorescent foci (Figure 4B).

We also examined the subcellular localization of endogenously expressed RGNEF in three mammalian cell lines by immunofluorescence microscopy. For these experiments we used HeLa cells (human), and the neuronal cell lines Neuro-2a and SN-56 (both mouse, Figure 4C). In HeLa cells, RGNEF (green) is found localized in both the nucleus (blue) and cytosol, appearing in a speckled pattern (Figure 4(Ci)). The same pattern is observed in partially differentiated Neuro-2a (Figure 4(Cii)) and SN-56 cells (Figure 4(Ciii)) where RGNEF is found in the nucleus and the cytosol and along the entire neurite extensions (Figure 4C, bottom row, inset), indicating that in the absence of any cellular stress and at endogenous expression levels RGNEF does not normally form large inclusions.

We next tested whether exposure to cellular stress modulates RGNEF misfolding and toxicity in yeast. We used MG132 (a proteasome inhibitor) or radicicol (an Hsp90 inhibitor) to induce protein quality control stress, hydrogen peroxide to induce oxidative stress, and tunicamycin or DTT to induce endoplasmic reticulum (ER) stress (Figure 5). Cells containing the indicated small molecules were evaluated by growth assays to assess RGNEF-dependent sensitivity or resistance to cellular stress. Figure 5A shows untreated control on non-inducing and inducing conditions (Figure 5(Ai,ii)). Figure 5B–E shows that the presence of MG132, hydrogen peroxide, and tunicamycin exacerbated the growth defect associated with full length wild type RGNEF expression (all *p* < 0.01).

To evaluate RGNEF inclusion formation in the presence of cellular stress, we employed mouse Neuro-2a cells and monitored the localization of endogenously expressed RGNEF by immunofluorescence microscopy. When Neuro-2a cells were treated with AZC (Figure 6ii), MG132 (Figure 6iii), or hydrogen peroxide (Figure 6v), RGNEF localization was unaltered compared to untreated cells (Figure 6i). In cells treated with radicicol, RGNEF aggregation appeared almost completely diffused (Figure 6iv). These results indicate the formation of large cytoplasmic inclusions might depend on its over-expression as noted before for other ALS proteins.

The split-ubiquitin system is an in vivo assay that detects protein–protein interaction in yeast by simple growth assays [30,31]. Using the split amino and carboxyl-terminal halves of ubiquitin fused to a bait and prey protein respectively, the split-ubiquitin assay detects both stable and transient interactions [30]. Furthermore, single amino acid residue substitutions within the NUb-bait fusions allow for additional assessment of the strength of the interaction [32] (Figure 7A). RUra3 serves as a reporter in the assay, where interactions between bait and prey fusions lead to the exposure of the degron (R) and mediates degradation of the Ura3 protein, which allows yeast to grow on media containing 5 FOA, but not on media lacking uracil as illustrated in Figure 7A.

In this study, RGNEF was used as the prey protein fused to the C-terminal half of ubiquitin (Figure 7A). A growth assay was performed to examine the proper expression of the RGNEF-CUb-RUra3 p (RGNEF-CRU) fusion constructs. DNAJB1-CUb-RUra3 p (DNAJB1-CRU) served as a non-toxic control. The split-ubiquitin constructs are under transcriptional control of the CUP1 promotor, whose activity increases with increasing Cu2+ ion concentrations in the media. The expression of RGNEF-CUb-RUra3 p is mildly toxic compared to DNAJB1-CUb-RUra3 p but both fusions allowed growth on media lacking uracil, which is a prerequisite for the split-ubiquitin assay (Figure 7B).

In a directed split-ubiquitin experiment we co-expressed RGNEF-CUb-RUra3 p with NUb-I/A/G-TDP-43, NUb-I/A/-RGNEF, or NUb-I/A/--PGK1. TDP-43 co-localizes with RGNEF in NCIs in ALS motor neurons and in fly models. Thus, TDP-43 served as a positive control for RGNEF interactors (Figure 7C, left). RGNEF was also tested for its capacity to form multimers. PGK1 is a human metabolic enzyme and is not expected to interact with RGNEF (negative control). Poor growth was observed on plates lacking uracil and growth was observed on 5 FOA plates when TDP-43 and RGNEF constructs were co-expressed, confirming that these two proteins interact. We also found that RGNEF can interact with other RGNEF molecules, albeit to a lesser extent than with TDP-43. PGK1 did not interact with RGNEF, indicating specificity for the TDP-43 and homomeric RGNEF interactions. These experiments show that the split-ubiquitin can be detected RGNEF interactions.

The RGNEF-CUb-RUra3 fusion protein was then screened against the human cDNA-based NUb-G fusion library and over 25 novel interactors with RGNEF were identified (Figure 7D). The largest functional group within these interactors were microtubule regulators, including the protein tau, which has been speculated to participate in ALS pathogenesis [33]. While the RGNEF contains a predicted microtubule-binding domain, this is, to our knowledge, the first experimental evidence of an interaction between RGNEF and the microtubule network. An additional protein associated with ALS, Sigmar-1, was also identified as an RGNEF interacting protein.

To further explore the novel finding that RGNEF interacts with the cellular microtubule network, we employed our yeast model and assessed genetic interaction between viable microtubule regulator gene deletions and RGNEF (Figure 8A). We found that deletion of several of these genes increased RGNEF toxicity. Kip2 is a kinesin-related motor protein involved in mitotic spindle positioning and stabilizing microtubules [34]. Expression of RGNEF in the absence of KIP2 enhanced RGNEF toxicity compared to wild type cells (Figure 8A,B, *p* < 0.05). ASE1 is a microtubule-associated family member (MAP) required for spindle elongation and stabilization [35]. RGNEF toxicity is also enhanced in yeast strains bearing deletions of ASE1 (Figure 8(Aiii,B), *p* < 0.05). VIK1 is a protein that forms kinesin-14 heterodimeric motor with KAR3 and localizes KAR3 at the mitotic spindle poles [36]. Expression of RGNEF in yeast cells bearing deletions for VIK1 (Figure 8(Aiv,B), *p* < 0.01) or KAR3 (Figure 8(Av,B), *p* < 0.01) was non-viable, indicating a strong synthetic toxicity and a strong genetic interaction. BEM3 is a Rho GTPase activating protein (RhoGAP) involved in actin cytoskeleton organization [37]. Deletion of this gene increases RGNEF toxicity to a lesser extent than the gene deletions tested above, indicating a stronger genetic interactions between RGNEF and microtubule-associated genes than with actin regulating genes (Figure 8(Av,B), *p* < 0.05).

Colchicine, which binds to monomeric tubulin and prevents its polymerization, and nocodazole, which binds to tubulin polymers and causes de-polymerization, were used to disrupt the microtubules [38,39]. Yeast cells grown on plates containing either small molecule show increased RGNEF toxicity (Figure 8(Ci,ii,D)). In Neuro-2a cells, treatment with colchicine and, to a lesser extent nocodazole, induce large cellular foci of endogenous RGNEF as shown by immunofluorescence microscopy (Figure 8E).

## 3. Discussion

RGNEF is a multidomain protein with a combination of a GEF and RNA binding domain that is unique within the human proteome, together with many other functional domains, yet its basic biological function is just starting to be clarified [20]. It has been implicated in a number of diseases, including cancer and neurodegenerative diseases, particularly in ALS [16,17,19,23,24,26,40,41,42]. RGNEF is similar to some of the other common ALS-associated proteins, such as TDP-43 and FUS, as it binds RNA and localizes to pathological NCIs [15,16,17,26]. For TDP-43, FUS and other ALS-associated proteins, there is a marked misfolding event that occurs in their prion-like domains, which either corresponds with, is caused by, or precipitates, their localization into NCI. Of note, RGNEF does not contain such a prion-like domain as defined before and the mechanisms underlying RGNEF misfolding and aggregation have been unclear [43]. Our data from a novel RGNEF yeast model and mammalian cells document that overexpressed RGNEF indeed can misfold and form cytosolic inclusions independent of other ALS proteins. We also found that RGNEF misfolding and inclusion formation is toxic to cells. Our genetic and split-ubiquitin results further confirm TDP-43 protein as interactors of RGNEF and shed light on a previously unknown function of RGNEF as a possible microtubule interactor and regulator.

To explore RGNEF, we established a novel yeast model. Similar studies in yeast have delivered profound insights into basic mechanisms of protein misfolding and the dysfunction of key cellular pathways associated with both normal cell function and disease, especially neurodegenerative disorders, such as ALS [44,45,46,47,48]. Compared to most other known ALS proteins, RGNEF is relatively understudied, possibly because RGNEF is challenging to explore as it is a large complex protein with many functional domains and extremely difficult to clone, express and purify. Our yeast model allows for relatively quick assessment of basic biological characteristics of RGNEF and its possible role in ALS.

Our experiments reveal that expression of wild type full length RGNEF is toxic in yeast. We further demonstrate that the GEF domain is required but not sufficient for this toxicity. This RGNEF toxicity pattern was confirmed in mammalian cells overexpressing RGNEF and its truncations. Analysis of RGNEF localization and aggregation reveals that wild type full length RGNEF and all truncations that contain the GEF and RNA binding domains create inclusions in the cytosol, which is a hallmark of misfolded proteins, such as TDP-43 and FUS. Our results thus indicate that RGNEF can form inclusions, plausibly because of misfolding, and can be toxic to cells. Our results point to a key role of the GEF domain in RGNEF toxicity but the exact mechanisms by which the individual domains contribute to RGNEF toxicity needs to be addressed in future experiments.

The predicted fALS-associated truncation of RGNEF (319) is toxic in yeast, whereas the leucine-rich domain alone is not toxic. The fALS-associated truncation 319 forms inclusions, indicating that the extended regions at the carboxy-terminus of the leucine-rich domain, which is mostly intrinsically disordered, drives aggregation (Appendix A). Of note, these RGNEF truncations have not yet been detected in post-mortem ALS tissue to date and their potential role in ALS pathogenesis remains unclear. Our results nevertheless indicate that RGNEF truncations can indeed be toxic in both yeast and mammalian cell models.

Protein folding stress in the endoplasmic reticulum (ER) is implicated in many neurodegenerative disorders, including ALS. Several regions of RGNEF are predicted to be intrinsically disordered, which can contribute to its misfolding and inclusion formation, particularly in environmental stress. These results demonstrate that ER stress exacerbates RGNEF toxicity suggesting that proper folding of RGNEF is required for proper function. Furthermore, our split-ubiquitin screen identified the Sigma receptor 1 (Sigmar1) as an RGNEF interactor. An ALS-associated mutation in the gene encoding Sigmar1 causes rapid aggregation of the protein in the ER leading to proteotoxic stress and impaired autophagy, accumulation of stress granules and cytoplasmic aggregation of the ALS proteins TDP-43, FUS, and Matrin3 [49]. While the precise mechanisms are still undefined, it is plausible that Sigmar1 links ER stress to RGNEF toxicity.

The split-ubiquitin screen and genetic and microscopic experiments identified microtubule regulators as the largest functional group of protein RGNEF interactors. ALS is characterized by the degeneration of motor neurons, their axons and synapses, highlighting the importance of maintaining the axonal transport systems built by microtubules [50,51]. Also, mutation of the microtubule subunit *TUB4 A* causes rare forms of fALS and there is evidence of microtubule dysfunction and collapse in both sALS and fALS [52]. The murine homolog of RGNEF, p190 RhoGEF, stabilizes the mRNA of *NEFL* (low molecular weight neurofilaments) [15]. This finding links the RNA binding capacity of RGNEF to the direct interaction with microtubule subunits and microtubule regulators. We identified TUB4 A as an RGNEF interactor. In addition, we found that genetic deletion of some microtubule regulators greatly modified RGNEF toxicity in the yeast model. Accordingly, RGNEF may interact with microtubules and regulate their function. We also found that small molecule-induced disruption of microtubules exacerbates RGNEF toxicity and treatment of Neuro-2a cells with nocodazole or colchicine induced the formation of RGNEF inclusions. Taken together, these findings suggest a functional link between RGNEF and the microtubule network that may be compromised by RGNEF misfolding and aggregation in ALS.

Collectively, our study indicates that RGNEF can contribute to cytotoxicity either alone or in combination with other ALS proteins, and impairing the proper function of the microtubule network. Future studies will need to clarify the underlying molecular mechanisms and the exact contributions of RGNEF and the smaller forms of RGNEF predicted to arise from ALS mutations to the pathogenesis of ALS.

## 4. Materials and Methods

### 4.1. Materials

All chemicals were purchased from Sigma-Aldrich (Saint Louis, MI, USA) and VWR. *S. cerevisiae* were grown on plates and in liquid media containing all essential nutrients aside from amino acids required to maintain plasmid selectivity, and 2% (*w*/*v*) glucose or galactose under induced and non-induced conditions [53].

All yeast strains are W303 Mat a or BY Mat α background and their derivatives. All gene deletion strains are in the BY Mat a background obtained from gene deletion library [54].

Plasmids for mammalian cell transfection were generated using Gateway cloning technology as previously described (Invitrogen, Carlsbad, CA, USA) [55]. Split-ubiquitin fusions are engineered by restriction digest and ligation-based cloning. DNA templates, pcDNA-RGNEF-myc and pcDNA-∆GEF-RGNEF-myc, for the generation of all constructs used in this study were generously provided by Dr. Cristian Droppelmann (London, ON, Canada).

HEK 293, Neuro-2a and SNC-34 cell lines were grown in DMEM (4 g/L glucose), or DMEM (1 g/L glucose) (ThermoFisher, Waltham, MA, USA) and supplemented with 10% bovine serum (Wisent, Saint-Jean-Baptiste, QC, Canada), Pen/strep (ThermoFisher), and 1% L-glutamine (ThermoFisher). Cells were detached with 5% trypsin (Gibco). All wash steps were carried out using cell culture grade PBS (ThermoFisher). Lipofectamine^®^ 2000 (Invitrogen) was used for transfections according to the manufacturer’s instructions.

### 4.2. Methods

#### 4.2.1. Yeast Transformations

Yeast strains were transformed using the LiAc procedure as previously described [56]. Cells were inoculated overnight in 3 mL YPD and allowed to grow to saturation. The following morning, cultures were switched to 27 mL YPD and allowed to grow to an OD600 of 0.4. Cells were washed twice in sterile water and suspended in 10 M LiAc solution for 30 min at 30 °C. Cells were then centrifuged and resuspended in LiAc, 50% PEG, DMSO, ss DNA carrier, and DNA and heat-shocked for 20 min at 42 °C. Then, cells were re-suspended in 1× Tris EDTA buffer and plated on selective agar plates (non-induced conditions). Transformation plates were placed at 30 °C for three days and colonies were chosen and streaked (4 biologicals/transformation) and allowed to grow for three days. Selection by amino acid exclusion was maintained.

#### 4.2.2. Spotting Assays

Yeast strains were inoculated overnight in 3 mL of selective growth media at 30 °C and grown to saturation. The optical density (OD600) was measured for each strain following 16 h of growth. Cultures were transferred to the top wells of sterile 96-well plates (650161, Greiner Bio-One, Kremsmuenster, Austria) and normalized to a starting OD600 of 0.1 in 200 uL. A five-fold serial dilution was performed using a multichannel pipette to transfer 30 uL of the normalized cultures into 120 uL of water. Cells were transferred to single agar plates using a sterilized 48-pin “replicate-plater”. Cells were spotted on non-inducing, overgrowth controls and inducing, selective plates. In the case of chemical treatments, cells were plated on agar plates containing the indicated concentration of various canonical stressors (50 uM MG132, 100 uM H202, 25 uM Radicicol, 1 ug/mL Tunicamycin, 5 mM DTT, 500 nM Colchicine, 500 nM Nocodazole).

#### 4.2.3. Quantification

For quantification, plates with yeast spotting assays were photographed on Bio-Rad GelDoc system (BioRad, Herculas, CA, USA). Images were first processed in Photoshop to remove color data and converted into black and white images. Images were then imported into ImageJ for quantification. A circular measuring tool was fit to the size of the third dilution and the white pixels were counted for each condition. Background was also measured and subtracted from each data point. Measured values were input into GraphPad prism to generate bar graphs. Statistical analysis was carried out on both normalized and unadjusted data applying One-way analysis of variance, Tukey post-hoc, error bars represent standard deviations.

#### 4.2.4. Split Ubiquitin Interaction Assay

Split-Ub expression clones were generated using standard restriction- and ligation-based cloning. RGNEF (bait) fused to the C-terminal half of ubiquitin and the Ura3 reporter was probed against a prey library fused to the N-terminus of ubiquitin. RGNEF, TDP-43, and PGK1 fused to the N-terminus of ubiquitin with different point mutations (isoleucine, alanine, glycine), were generated and used in directed screens. Strains were generated expressing both the bait and prey constructs and plated on plates containing 500 µM CuSO_4_ and 5-FOA (positive selection media), and plates containing uracil and 5-FOA (negative selection media), and on YPD (overgrowth control). Colonies were isolated and DNA was extracted following overnight incubation in selective media using the Zymoprep yeast plasmid miniprep kit (ZYMO, Irvine, CA, USA). DNA was sequenced using primers against the N-terminus of ubiquitin. Genes were identified by sequence comparison using BLAST.

#### 4.2.5. Microscopy

Yeast strains were inoculated in 3 mL of non-inducing media and allowed to grow overnight at 30 °C. The following evening, cultures were spun and washed twice in water, then resuspended in inducing growth media and returned to 30 °C for overnight growth. Following a minimum of 16 h induction, yeast cells were pipetted onto glass slides and sealed under coverslips using nail-polish. Fluorescence microscopy was performed using the Cytation5 cell imaging multi-mode reader (BioTek, Winooski, WI, USA). The following LED cubes and imaging filter cubes from biotech were employed for GFP tagged proteins: 465 LED 1225001 Rev J, GFP 469/525 1225101 Rev J, BioTek; and for CFP-tagged proteins: 465 LED 1225001 Rev J, CFP 445/510 1225107 Rev I. Images were captured on the 20× objective. Image analysis was completed using the Gen5 imaging software (BioTek).

Yeast microscopy data were quantified by taking three biological fields on the 20× objective and overlaying a grid to cut the image into four quadrants. Total of 50 cells were counted and sorted as either showing a diffuse signal or containing one or more fluorescent foci for each biological field. Stacked bar graphs with SD error bars were generated in GraphPad Prism using this data.

#### 4.2.6. Mammalian Cell Culture

Mammalian cells used include human (HeLa, HEK 293), and mouse-derived (Neuro-2a, SN-56) lines. All lines were grown in DMEM, 5% bovine serum in the presence of pen/strep. All cells were cultured at 37 °C in the presence of 5% CO2. Partial differentiation of Neuro-2a and SN-56 cells was carried out by growing cells for three days in DMEM in the absence of bovine serum. Transfection of pcDNA 3.1 GFP RGENF WT, ∆Cterm and ∆Leu was carried out with Lipofectamine^®^ 2000 as described previously (Invitrogen 11668030) in Opti-MEM™ reduced serum media (31985062, Thermo Fischer).

#### 4.2.7. Immunofluorescence Microscopy

HEK293 cells were seeded at 1250 cells/well in 96-white well plates and transfected as stated above. The Cell Titer Glo 2.0 (Promega, Madison, WI, USA), and the luciferase-based luminescence viability assay was used to detect viable cells. Luminescence was measured on Cytation5 plate reader (BioTek).

Prior to imaging, cells were seeded at 2500 cells/well in Nunc, 8-well chamber slides in DMEM, 5% or 1% bovine serum. Media was removed and replaced with growth medium containing the indicated chemical concentrations. Control wells were replaced with the same growth media lacking any additional treatment. Following treatment, media was aspirated and cells were probed with anti-RGNEF (ab122399, Abcam, Cambridge, UK), and anti-Tubulin (Abcam ab4074) overnight. Primary antibody solution was aspirated and cells were washed in PBS and incubated with Alexa Fluor 488 and Alexa Fluor 688 secondary antibodies (ThermoFisher) for one hour. Cells were fixed to the slide using 30% paraformaldehyde. ProLong Diamond antifade mounting solution (Thermo-Fischer Scientific, P36965), was added to fixed cells and covered by a coverslip. Slides were sealed with clear nail polish.

Fluorescence microscopy was performed using the Cytation5 cell imaging multi-mode reader (BioTek). The following LED cubes and imaging filter cubes from biotech were employed for RGNEF probed with Alexa Fluor 488 secondary: 465 LED 1225001 Rev J, GFP 469/525 1225101 Rev J, BioTek); for tubulin probed with Alexa Fluor 688: TEXAS RED 586/647, 1225 102 Rev H. Images were captured on the 20× objective. Image analysis was completed using the Gen5 imaging software (BioTek).

## 5. Summary Statement

We have characterized a novel yeast model expressing human RGNEF, a protein implicated in amyotrophic lateral sclerosis. Using this yeast model and mammalian cells we have shown that overexpressed RGNEF forms inclusions, is cytotoxic, and is sensitive to cellular stress. We also uncovered a potential role for RGNEF as a microtubule regulator.

## Figures and Tables

**Figure 1 ijms-21-05597-f001:**
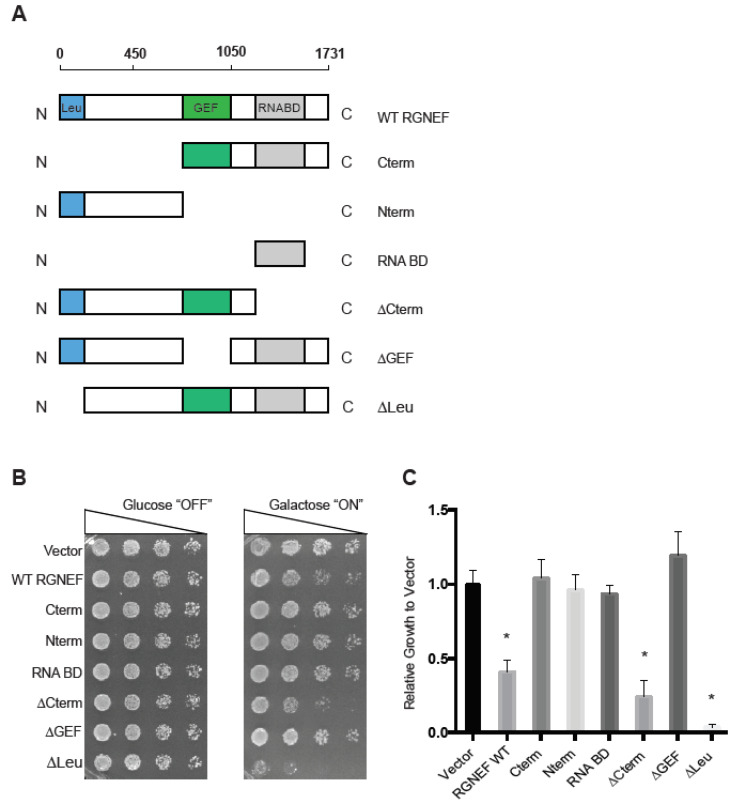
Rho guanine nucleotide exchange factor (RGNEF) toxicity in yeast. (**A**) RGNEF WT (full length, wild type) and domain-deleted truncations are illustrated. (**B**) Yeast growth assay of RGNEF WT and mutants. Cells were plated on non-inducing and inducing agar plates in one to five serial dilutions to observe the growth phenotype associated with expressing each variant (left). (**C**) Results as shown in B were quantified and normalized to growth of the control cells (vector). Error bars represent standard deviations. One way analysis of variance, Tukey post hoc show statistical significance (*) of reduced growth for RGNEF WT (*p* < 0.05), ∆Cterm (*p* < 0.05), and ∆Leu (*p* < 0.01) compared to controls.

**Figure 2 ijms-21-05597-f002:**
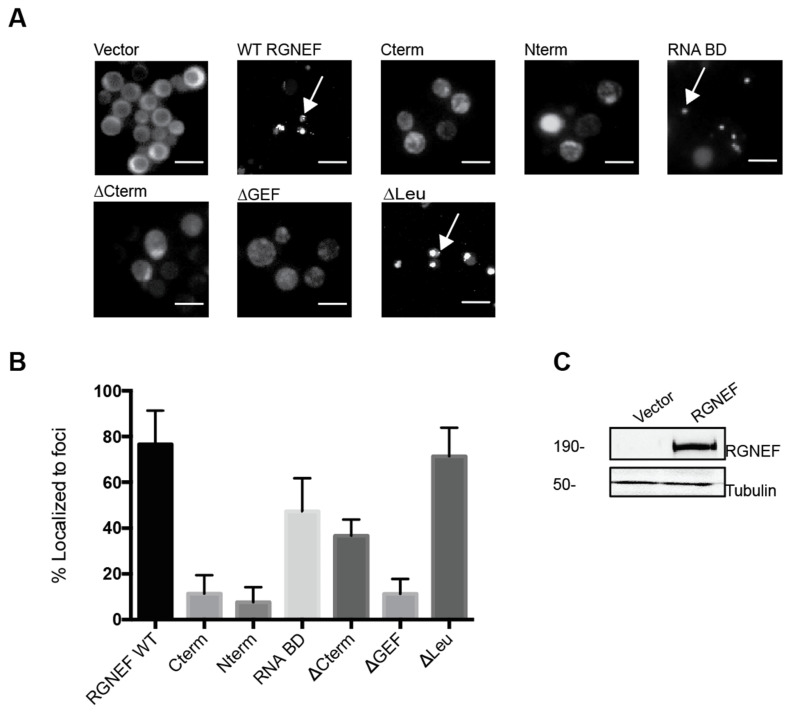
RGNEF-YFP localization in yeast. (**A**) RGNEF-YFP (wild type full length and truncated variants) localization and inclusion formation was observed using fluorescence microscopy in live yeast cells. Arrows indicate inclusions. The scale bar corresponds to 10 μm (**B**) Bar graph represents the quantification of data as shown in A as percentile of cell containing inclusions. (**C**) RGNEF expression is shown by Western blot analysis. Error bars represent standard deviations.

**Figure 3 ijms-21-05597-f003:**
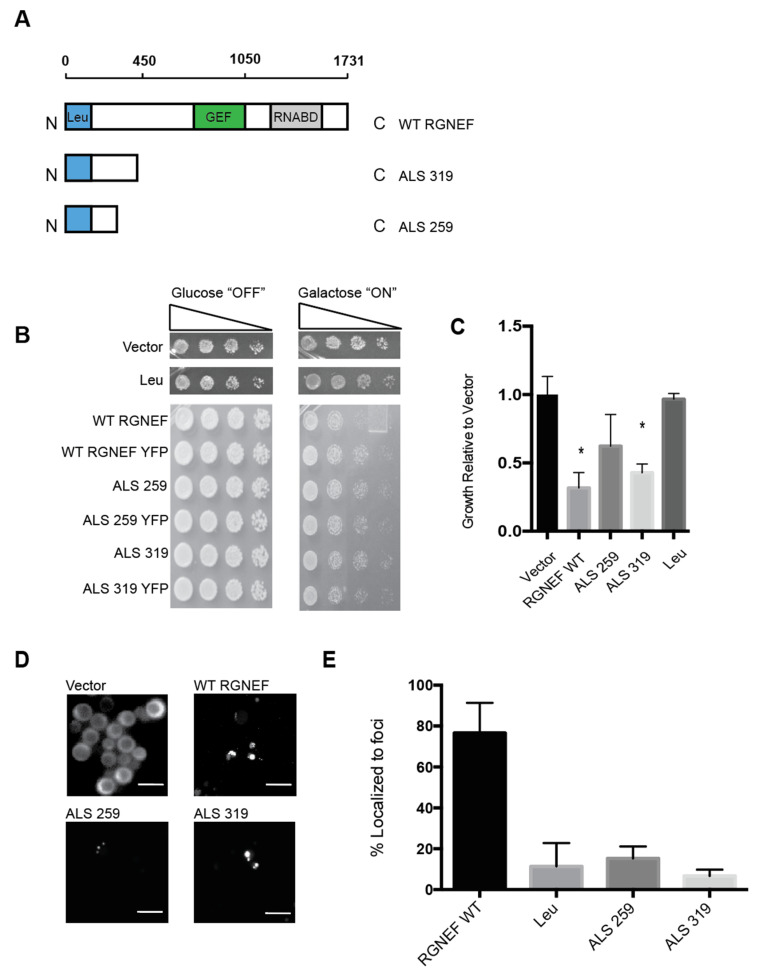
Yeast model of predicted amyotrophic lateral sclerosis (ALS)-associated RGNEF variants. (**A**) Schematic presentation of full length wild type RGNEF and truncated ALS-associated constructs. The truncations are named according to their predicted amino acids lengths. (**B**) Yeast growth assay of RGNEF WT and predicted truncations. (**C**) Quantification of data as shown in B. * indicates statistical significance (*p* < 0.05). (**D**) Localization predicted ALS-associated RGNEF variants fused to YFP by fluorescence microscopy. The scale bar corresponds to 10 μm. (**E**) Quantification of the fluorescent microscopy as show in D as percentages of cells containing fluorescent foci. Error bars represent standard deviations.

**Figure 4 ijms-21-05597-f004:**
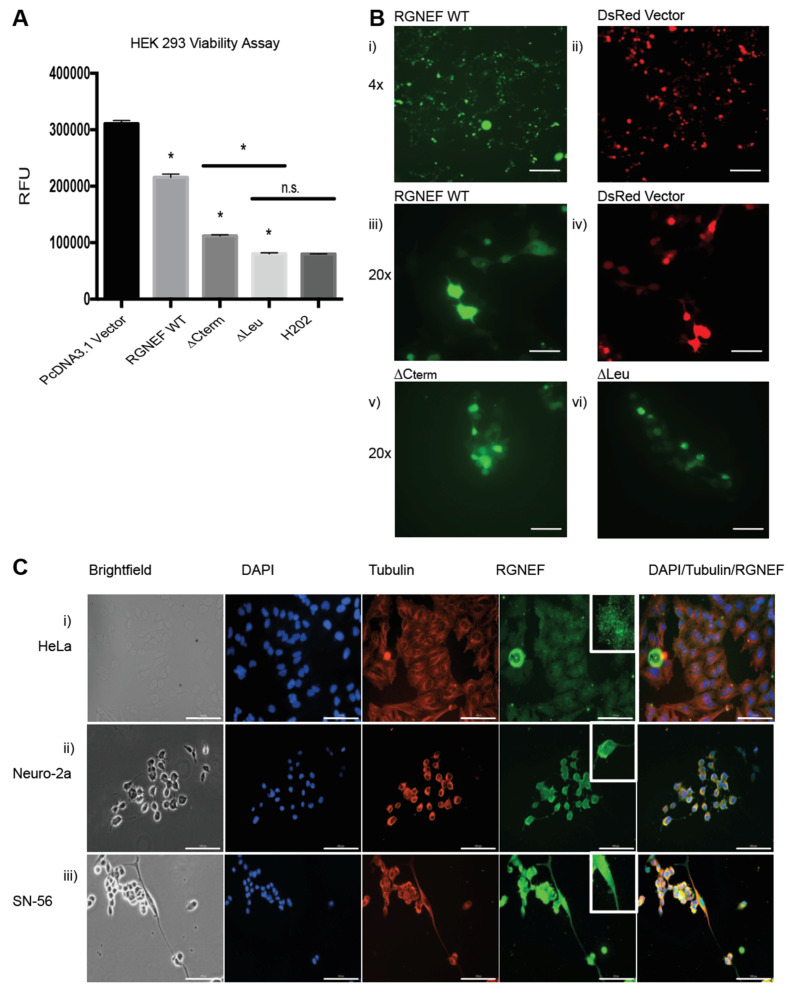
RGNEF toxicity and localization in mammalian cells. (**A**) Transfection of HEK 293 cells with DsRED control and RGNEF WT, ∆Cterm, and ∆Leu constructs. Following transfection, cell viability was measured by luciferase assay and quantified. * indicates statistical significance (*p* < 0.05) and n.s. indicates not statistically significant compared to control. (**B**) Fluorescence microscopy of GFP-tagged RGNEF constructs transfected into HEK 293, (i) full length wild type RGNEF and (ii) DsRed vector control (4× magnification, scale bar = 500 µm), (iii) RGNEF WT (iv) DsRed expressing vector control, ∆Cterm overexpression and (vi) ∆Leu overexpression (20× magnification, scale bar = 100 µm). (**C**) Endogenous RGNEF localization is monitored by immunofluorescence microscopy in HeLa (i), Neuro-2a (ii), and SN-56 cells (iii) by immune fluorescence microscopy (20× magnification, scale bar = 100 µm). Error bars represent standard deviations.

**Figure 5 ijms-21-05597-f005:**
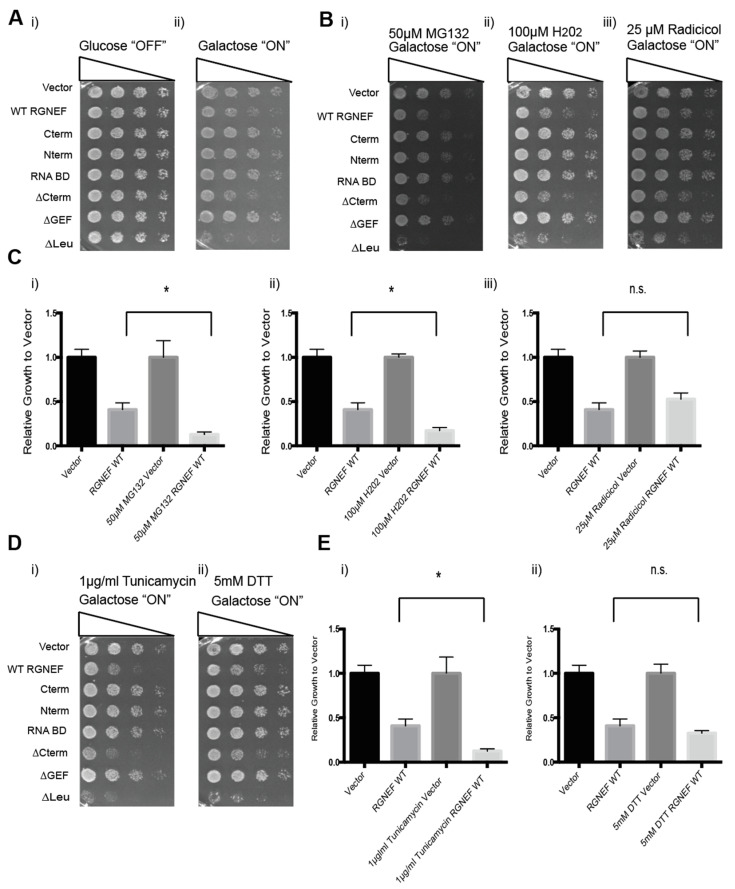
RGNEF toxicity under cellular stress. (**A**) Yeast growth assay of yeast cells under non-inducing (i) and inducing conditions (ii) for the expression of full length wild type and truncated RGNEF variants. (**B**) Yeast growth assay of cells expressing full length wild type and truncated RGNEF variants under inducing conditions containing (i) 50 µM MG132, (ii) 100 µM H202, and (iii) 25 µM Radicicol. (**C**) Quantification of the growth assays as show in B normalized data to the vector control, (i) 50 µM MG132, (ii) 100 µM H202, and (iii) 25 µM Radicicol. (**D**) Yeast growth assay of cells expressing full length wild type and truncated RGNEF variants containing (i) 0.1 µg/mL tunicamycin, and (ii) 5 mM DTT. (**E**) Quantification of the growth assay as shown in D normalized data to vector controls, (i) 1 µg/mL Tunicamycin and (ii) 5 mM DTT. Error bars represent standard deviations. * indicates statistical significance (*p* < 0.05) and n.s. indicates not statistically significant compared to vector control.

**Figure 6 ijms-21-05597-f006:**
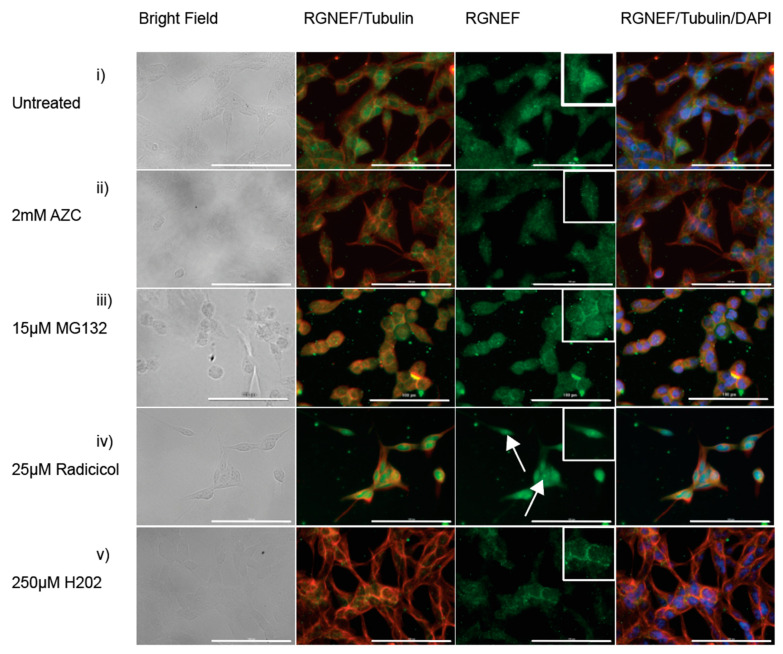
RGNEF localization under cellular stress in Neuro-2a cells. Endogenous RGNEF is visualized by immuno-fluorescence microscopy in untreated and stress treated conditions in semi-differentiated Neuro-2a cells, (**i**) untreated, (**ii**) 2 mM AZC, (**iii**) 15 µM MG132, (**iv**) 25 µM radicicol (white arrows indicate nuclear RGNEF inclusions), and (**v**) 250 µM hydrogen peroxide. (40× magnification, scale bar = 100 µm).

**Figure 7 ijms-21-05597-f007:**
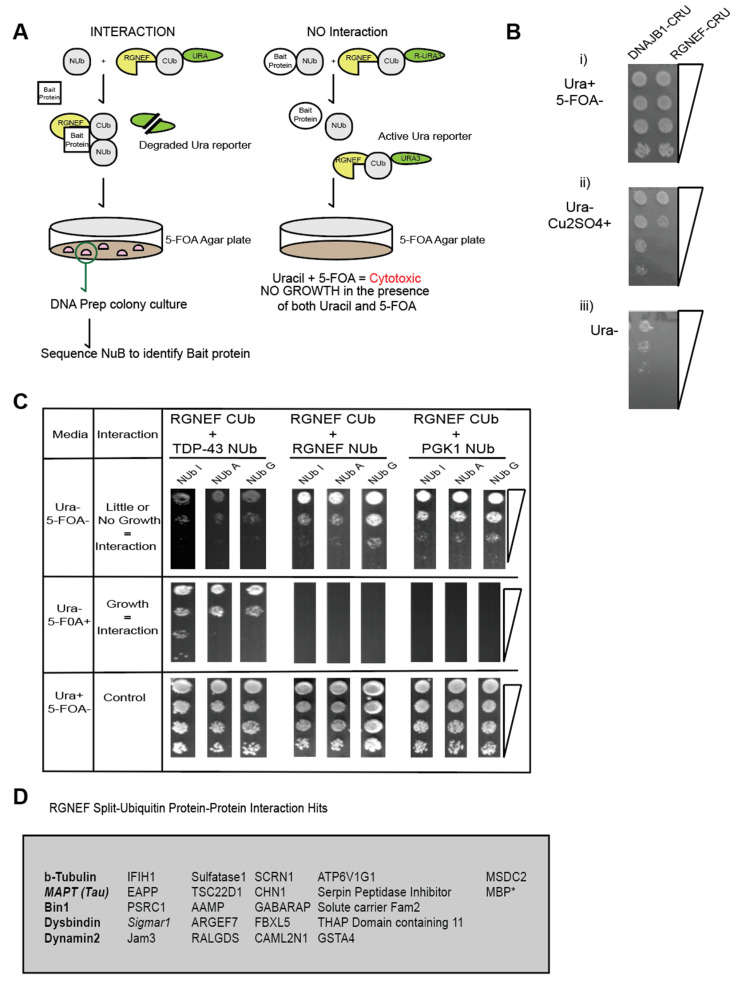
(**A**) RGNEF Split-Ub screen identifies microtubule regulators as interactors with RGNEF. (**B**) Growth assays of yeast cells expressing DNAJB1-CRU (control) and RGNEF-CRU on inducing and non- inducing media, (i) media containing uracil (Ura+) but not containing 5FOA (5FOA-), (ii) no uracil (Ura-) and copper sulfate (Cu2SO4), and (iii) no uracil (Ura-). (**C**) Split-Ub interactions assays with RGNEF-CRU and TDP-43-NUb (left,) RGNEF-NUb (middle), and PGK1-NUb (right). (**D**) List of all candidate RGNEF interactors identified in a split-Ub screen using RGNEF-CUb-CRU as prey and a human NUbG library as bait.

**Figure 8 ijms-21-05597-f008:**
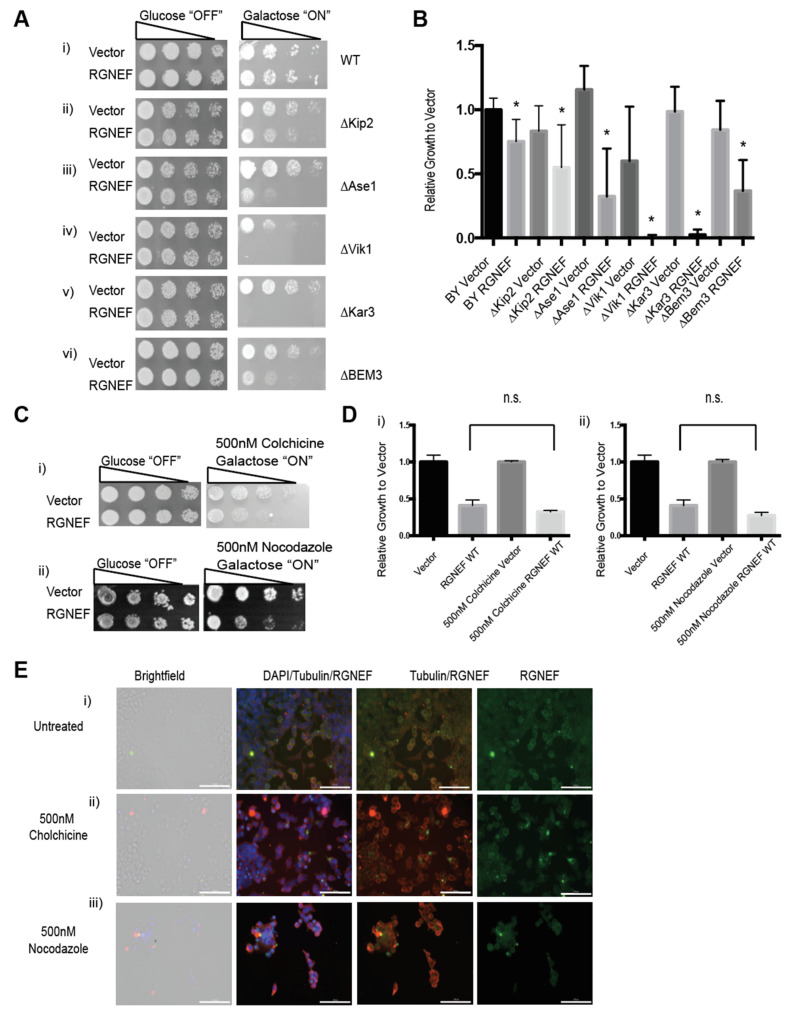
Genetic interaction between RGNEF and microtubule-associated genes. (**A**) Growth assay of yeast cells expressing full length wild type RGNEF in a wild type yeast strain and microtubule-associated gene deletion strains: (i) WT strain, (ii) ∆Kip2, (iii) ∆Ase1, (iv) ∆Vik1, (v) Kar3, and (vi) ∆BEM. (**B**) Quantification of growth assay as shown in A. Error bars represent standard deviations. * indicates statistical significance (*p* < 0.05) compared to vector control. (**C**) Growth assay of yeast cells expressing wild type RGNEF in a wild type yeast strain grown on inducing media containing (i) 500 nM Colchicine and (ii) 500 nM Nocodazole. (**D**) Quantification of growth assays as shown in C. Error bars represent standard deviations, n.s. indicates not statistically different. (**E**) Immunofluorescence microscopy of partially differentiated Neuro-2a cells detecting endogenous RGNEF: (i) untreated cells, (ii) 500 nM colchicine, and (iii) 500 nM Nocodazole. Error bars represent standard deviations. (20× magnification, scale bar = 100 µm).

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
