# Peer review of "Inclusion Formation and Toxicity of the ALS Protein RGNEF and Its Association with the Microtubule Network"

_ijms, 2020, doi:10.3390/ijms21165597_

Round 1
Reviewer 1 Report
In this study, the authors aimed to elucidate the contribution of RGNEF to the pathological mechanism of ALS, and revealed that overexpression of RGNEF promotes the formation of inclusions that exert toxicity. In addition, the authors also revealed that RGNEF is involved in microtubule regulation. However, the presented results lack consistency. For instance, TDP-43, an ALS-associated protein, interacts with RGNEF, even thoughthe authors conclude that RGNEF can form cytosolic inclusions independent of ALS-associated proteins (Line. 285). Meanwhile, although GEF domain of RGNEF seems to play a key role in its toxicity (Figure 1 and 2), why truncated mutants lacking GEF domain in Figure 3 are toxic? Moreover, the presented results do not show a substantial relationships among the toxicity, the formation of inclusions, and the regulation of microtubules mediated by RGNEF, and thereby the underlying mechanisms of howRGNEF exerts cytotoxicity are ambiguous. Thus,this study is too preliminary for publication in IJMS.
Author Response
In this study, the authors aimed to elucidate the contribution of RGNEF to the pathological mechanism of ALS, and revealed that overexpression of RGNEF promotes the formation of inclusions that exert toxicity. In addition, the authors also revealed that RGNEF is involved in microtubule regulation.
We thank the reviewer for their critical and thoughtful comments.
However, the presented results lack consistency. For instance, TDP-43, an ALS-associated protein, interacts with RGNEF, even thoughthe authors conclude that RGNEF can form cytosolic inclusions independent of ALS-associated proteins (Line. 285).
We indeed find that RGNEF independently forms inclusions in our yeast model. Of note, yeast do not express homologues of TDP-43 or other ALS proteins (e.g. FUS, C9orf72), which excludes the possibility that RGNEF can only form inclusions when co-expressed with these other proteins. Also, our mammalian cell models express TDP-43 but only endogenous levels (not over expressed), which do not form cytosolic inclusions under the experimental conditions we tested here. To clarify this, we added the term "over expressed" to line 285.
Meanwhile, although GEF domain of RGNEF seems to play a key role in its toxicity (Figure 1 and 2), why truncated mutants lacking GEF domain in Figure 3 are toxic?
This seems to be a misunderstanding. Indeed, we show that the GEF domain of RGNEF is not required and also not sufficient for toxicity (Figure 1). The context of the truncations we used is clearly important. We thus cannot detect any inconsistency.
Also, we argue that the toxicity and inclusion formation of the relatively small RGNEF fragment ALS 319 is mostly governed by its intrinsically disordered domain (Figure 3 and supplmentary figure and Discussion). Of note, the deltaC construct contains this intrinsically disordered domain. Again, there is not inconsistency.
Moreover, the presented results do not show a substantial relationships among the toxicity, the formation of inclusions, and the regulation of microtubules mediated by RGNEF, and thereby the underlying mechanisms of howRGNEF exerts cytotoxicity are ambiguous.
We (and the other two reviewers) respectfully disagree. Our data show that over expressing RGNEF in yeast and mammalian cells is toxic and leads to the formation of inclusions, i.e. inclusion formation and toxicity correlate. Showing a direct mechanistic link between inclusions and toxicity is a major and complex problem in this field for many different proteins and diseases and will keep researchers busy for years to come.
We also provide evidence for protein-protein interactions (split-ub, Figure 7) and genetic interactions (Figure 8) between RGNEF, the microtubuli network, and RGNEF toxicity and inclusion formation. We also phrased this connection very carefully in our Dicussion (e.g. lines 336-340).
Thus,this study is too preliminary for publication in IJMS.
We are convinced that our work contributes significantly to a better understanding on how RGNEF might contribute to ALS and identified its capacity to form inclusions and its interactions with the microtubuli network as mechanistic links.
Reviewer 2 Report
In this article, authors have investigated the role of RGNEF to ALS pathogenesis. They further found that RGNEF is toxic when overexpressed and form inclusions. They have shown that fALS-associated mutation in ARGHEF gives rise to an inclusion-forming and toxic protein. Authors suggest a possible mechanism by which RGNEF misfolding and toxicity may cause impairment of the microtubule network and contribute to ALS pathogenesis.
The following points should be addressed.
- line 36, Should be Fused in Sarcoma. Some spelling mistakes need to be proofread.
- Line 99, Is that description in this line matches with the figure 1A, ^Cterm? Overall all the figures quality could be improved.
- Is the effect of ALS formation being dependent on the TDP43 interaction or Microtubule regulation?
- Line 312, Is there any report about full-length RGNEF overexpression in ALS tissues?
- As with figure 1A, authors could scan the protein sequence for the IDPs from available bioinformatics tools.
- Line 136: There is no figure 3F.
Author Response
We thank the reviewer for their thoughtful and thorough comments:
1. line 36, Should be Fused in Sarcoma. Some spelling mistakes need to be proofread.
We corrected this typographical error and all others.
2. Line 99, Is that description in this line matches with the figure 1A, ^Cterm? Overall all the figures quality could be improved.
We corrected this error and the description in the text does now match the figure (Figure 1).
3. Is the effect of ALS formation being dependent on the TDP43 interaction or Microtubule regulation?
Yes, over expressed RGNEF can form inclusions independent of TDP-43 or dysregulated microtubles. We revised the discussion to clarify this point (e.g. line 285.
4. Line 312, Is there any report about full-length RGNEF overexpression in ALS tissues?
We are not aware of any report that documents over expression of RGNEF in ALS. We simply use RGNEF over expression in our yeast and mammalian cell model in analogy to many other misfolded proteins (e.g. TDP-43, FUS) to produce a tractable experimental system.
5. As with figure 1A, authors could scan the protein sequence for the IDPs from available bioinformatics tools.
ID domains in RGNEF are have been identified using three different algorithms. This is shownm in Supplementary Figure 1.
6. Line 136: There is no figure 3F.
We corrected this (now Figure 3D and E).
Reviewer 3 Report
A sound paper which establishes a potential role for a guanine nucleotide exchange factor in ALS. The manuscript is well written with occasional typos which can be addressed in the drafting. Evidence is provided for this although the small number of cases where a mutation has been found suggest that it is a little early to pronounce it having a major effect, l314.
Author Response
We thank the reviewer for their positive assessment of our work.
We also carefully searched for typographical errors and corrected them.
Round 2
Reviewer 1 Report
We indeed find that RGNEF independently forms inclusions in our yeast model. Of note, yeast do not express homologues of TDP-43 or other ALS proteins (e.g. FUS, C9orf72), which excludes the possibility that RGNEF can only form inclusions when co-expressed with these other proteins. Also, our mammalian cell models express TDP-43 but only endogenous levels (not over expressed), which do not form cytosolic inclusions under the experimental conditions we tested here. To clarify this, we added the term "over expressed" to line 285.
Response:
Over expressed RGNEF forms inclusions without other ALS proteins, and with TDP-43 in yeast and mammalian cells, respectively. The reviewer undetstood the authors’ claime on the line 285. However, the reviewers’ question is that RGNEF can forms inclusions without requiring TDP-43 in mammalian cells. To this end, the authors should test whether RGNEF forms inclusions in TDP-43 knockout (or knockdown) cells. If yes, the consistency of results are demonstrated. If not, the authors should discuss difference between yeast and mammalinans.
Meanwhile, although GEF domain of RGNEF seems to play a key role in its toxicity (Figure 1 and 2), why truncated mutants lacking GEF domain in Figure 3 are toxic?
This seems to be a misunderstanding. Indeed, we show that the GEF domain of RGNEF is not required and also not sufficient for toxicity (Figure 1). The context of the truncations we used is clearly important. We thus cannot detect any inconsistency.
Also, we argue that the toxicity and inclusion formation of the relatively small RGNEF fragment ALS 319 is mostly governed by its intrinsically disordered domain (Figure 3 and supplmentary figure and Discussion). Of note, the deltaC construct contains this intrinsically disordered domain. Again, there is not inconsistency.
Response:
The reviewer undetstand the authors’ claim that“we show that the GEF domain of RGNEF is not required and also not sufficient for toxicity”. Actually, the authors claim is correct but this sentence means that the authors did not come to any conclusion to gain a better understanding of the function of RGNEF. The reviewer felt that the experiments do not make sense.
Moreover, the presented results do not show a substantial relationships among the toxicity, the formation of inclusions, and the regulation of microtubules mediated by RGNEF, and thereby the underlying mechanisms of how RGNEF exerts cytotoxicity are ambiguous.
We (and the other two reviewers) respectfully disagree. Our data show that over expressing RGNEF in yeast and mammalian cells is toxic and leads to the formation of inclusions, i.e. inclusion formation and toxicity correlate. Showing a direct mechanistic link between inclusions and toxicity is a major and complex problem in this field for many different proteins and diseases and will keep researchers busy for years to come.
Response:
The reviewer undetsstand the authors’ claim that “Our data show that over expressing RGNEF in yeast and mammalian cells is toxic and leads to the formation of inclusions”. However, since the construct that forms inclusions and that exerts toxicity are not consistent, the involvement of the inclusion in its toxicity cannot be claimed. The authors should explain why the RNA BD can not show toxicity, yet it forms inclusions. In addition, the delta Ctem can show toxicity but not evidently forms inclusions.
We also provide evidence for protein-protein interactions (split-ub, Figure 7) and genetic interactions (Figure 8) between RGNEF, the microtubuli network, and RGNEF toxicity and inclusion formation. We also phrased this connection very carefully in our Dicussion (e.g. lines 336-340).
Thus,this study is too preliminary for publication in IJMS.
We are convinced that our work contributes significantly to a better understanding on how RGNEF might contribute to ALS and identified its capacity to form inclusions and its interactions with the microtubuli network as mechanistic links.
Response:
On the lines 336-340, the authors mentioned following 3 points;
1. RGNEF interactes with the reguratory proteins of microtubuli network.
2. Kockout of the proteins enhances RGNEF toxicity.
3. The microtubuli damage induces RGNEF aggregation.
These sentences are just bullet points of the results but not discussion. The authors should discuss the implications of these findings, and show proposal models to make easy to understand the interpritation of the results.
Additional minor points;
1. In line 57, “an” should be “a” and three is an extra “(”.
2. In the legend of Figure 2. A The explanation about white arrows is lacking.
3. In line 121, “2c” should be “2C”.
4. In line 136, “Figure 3 E and F” should be corrected to “Figure 3D and E”.
5. In line “a lesser extent then the gene deletions” should be corrected to “a lesser extent than the gene deletions”.
6. “ander” should be “under”.
7. “Raficicol” should be “Radicicol”.
Finally, minor but serious point;
Panel of Fig 1B is dupulicated in Fig 5A, which may be undesirable.
Author Response
The responses to the reviewers comments can be found below.
Over expressed RGNEF forms inclusions without other ALS proteins, and with TDP-43 in yeast and mammalian cells, respectively. The reviewer undetstood the authors’ claime on the line 285. However, the reviewers’ question is that RGNEF can forms inclusions without requiring TDP-43 in mammalian cells. To this end, the authors should test whether RGNEF forms inclusions in TDP-43 knockout (or knockdown) cells. If yes, the consistency of results are demonstrated. If not, the authors should discuss difference between yeast and mammalinans.
TDP-43 knockouts are lethal and knockdowns highly toxic in mammalian cells. Therefore, such as experiment will not give any useful insights into RGNEF inclusion formation. Also, under our experimental conditions, TDP-43 does not form inclusions and is localized to the nucleus. It thus seems highly unlikely that TDP-43 would seed RGNEF inclusion formation in the cytoplasm of mammalian cells.
The reviewer undetstand the authors’ claim that“we show that the GEF domain of RGNEF is not required and also not sufficient for toxicity”. Actually, the authors claim is correct but this sentence means that the authors did not come to any conclusion to gain a better understanding of the function of RGNEF. The reviewer felt that the experiments do not make sense.
Our structure/function analysis regarding RGNEF inclusion formation and toxicity aimed at determining whether specific predicted functional domains contribute to inclusion formation and toxicity. Our results demonstrate, as previously reported for many other misfolded proteins (e.g. TDP-43 and FUS), that there is no single functional domain within RGNEF that governs inclusion formation or toxicity. Our further results indicate that the intrinsically disordered domain at the amino terminus may drive inclusion formation and toxicity. This characterization of RGNEF is novel and indicated that the full length RGNEF protein is key to understanding its toxicity and inclusion formation.
The reviewer undetsstand the authors’ claim that “Our data show that over expressing RGNEF in yeast and mammalian cells is toxic and leads to the formation of inclusions”. However, since the construct that forms inclusions and that exerts toxicity are not consistent, the involvement of the inclusion in its toxicity cannot be claimed. The authors should explain why the RNA BD can not show toxicity, yet it forms inclusions. In addition, the delta Ctem can show toxicity but not evidently forms inclusions.
As established for many other misfolded proteins (e.g. TDP-43 and FUS) there is no clear correlation between RGNEF inclusion formation and toxicity. In fact, this is part of an ongoing debate about the role of inclusions in (neuro)toxicity. The overly simplistic view that inclusions are always toxic has been disproven for many proteins in many diseases, including ALS. Our results again demonstrate for the first time that full length RGNEF is toxic and forms inclusions when over expressed and we do not make any claims regarding a correlation or contributions of individual domains.
Response:
On the lines 336-340, the authors mentioned following 3 points;
1. RGNEF interactes with the reguratory proteins of microtubuli network.
2. Kockout of the proteins enhances RGNEF toxicity.
3. The microtubuli damage induces RGNEF aggregation.
These sentences are just bullet points of the results but not discussion. The authors should discuss the implications of these findings, and show proposal models to make easy to understand the interpritation of the results.
We list our major results to support our notion that RGNEF genetically and physically interacts with the microtuble network and that RGNEF misfolding and toxicity is modulated by the microtuble network. These findings indicate a previously unknown link between RGNEF and microtubles. Our data do not yet allow to propose a more detailed mechanistic model as many different scenarios are possible and we did not want to over-interpret our data.
Additional minor points;
1. In line 57, “an” should be “a” and three is an extra “(”.
2. In the legend of Figure 2. A The explanation about white arrows is lacking.
3. In line 121, “2c” should be “2C”.
4. In line 136, “Figure 3 E and F” should be corrected to “Figure 3D and E”.
5. In line “a lesser extent then the gene deletions” should be corrected to “a lesser extent than the gene deletions”.
6. “ander” should be “under”.
7. “Raficicol” should be “Radicicol”.
Thanks, we fixed all these typos.
Finally, minor but serious point;
Panel of Fig 1B is dupulicated in Fig 5A, which may be undesirable.
The pictures are indeed identical since they are from the same series of experiments (and used to be in one giant figure). We have many biological repeats of these experiments and replaced Figure 5A with one of these repeats in the revised version of our manuscript to avoid confusion.
Reviewer 2 Report
Authors have provided a satisfactory response to concerns. However, the consistency of the presentation of data is missing, which needs to be addressed before publications. Authors may use available software for example GraphPad Prism or similar to make these graphs with more clarity.
Author Response
We thank the reviewer for their time and their thorough evaluation of our manuscript.
Round 3
Reviewer 1 Report
TDP-43 knockouts are lethal and knockdowns highly toxic in mammalian cells. Therefore, such as experiment will not give any useful insights into RGNEF inclusion formation. Also, under our experimental conditions, TDP-43 does not form inclusions and is localized to the nucleus. It thus seems highly unlikely that TDP-43 would seed RGNEF inclusion formation in the cytoplasm of mammalian cells.
Response: the review understood the situation.
Our structure/function analysis regarding RGNEF inclusion formation and toxicity aimed at determining whether specific predicted functional domains contribute to inclusion formation and toxicity. Our results demonstrate, as previously reported for many other misfolded proteins (e.g. TDP-43 and FUS), that there is no single functional domain within RGNEF that governs inclusion formation or toxicity. Our further results indicate that the intrinsically disordered domain at the amino terminus may drive inclusion formation and toxicity. This characterization of RGNEF is novel and indicated that the full length RGNEF protein is key to understanding its toxicity and inclusion formation.
As established for many other misfolded proteins (e.g. TDP-43 and FUS) there is no clear correlation between RGNEF inclusion formation and toxicity. In fact, this is part of an ongoing debate about the role of inclusions in (neuro)toxicity. The overly simplistic view that inclusions are always toxic has been disproven for many proteins in many diseases, including ALS. Our results again demonstrate for the first time that full length RGNEF is toxic and forms inclusions when over expressed and we do not make any claims regarding a correlation or contributions of individual domains.
Response: To summarize the authors' comments, the only real novelty here is that overexpressed RGNEF forms the toxic inclusion. Therefore, it’s not clear just how much these findings move the field forward. The reviewer believe that this study is preliminary and appears to make marginal advances in this research area.